# More Space, Less Noise—New-generation Low-Field Magnetic Resonance Imaging Systems Can Improve Patient Comfort: A Prospective 0.55T–1.5T-Scanner Comparison

**DOI:** 10.3390/jcm11226705

**Published:** 2022-11-12

**Authors:** Thilo Rusche, Jan Vosshenrich, David J. Winkel, Ricardo Donners, Martin Segeroth, Michael Bach, Elmar M. Merkle, Hanns-Christian Breit

**Affiliations:** Department of Radiology, Clinic of Radiology & Nuclear Medicine, University Hospital Basel, 4031 Basel, Switzerland

**Keywords:** low-field MRI, scanner-comparison, patient comfort, MRI

## Abstract

Objectives: The objectives of this study were to assess patient comfort when imaged on a newly introduced 0.55T low-field magnetic resonance (MR) scanner system with a wider bore opening compared to a conventional 1.5T MR scanner system. Materials and Methods: In this prospective study, fifty patients (mean age: 66.2 ± 17.0 years, 22 females, 28 males) underwent subsequent magnetic resonance imaging (MRI) examinations with matched imaging protocols at 0.55T (MAGNETOM FreeMax, Siemens Healthineers; Erlangen, Germany) and 1.5T (MAGNETOM Avanto Fit, Siemens Healthineers; Erlangen, Germany) on the same day. MRI performed between 05/2021 and 07/2021 was included for analysis. The 0.55T MRI system had a bore opening of 80 cm, while the bore diameter of the 1.5T scanner system was 60 cm. Four patient groups were defined by imaged body regions: (1) cranial or cervical spine MRI using a head/neck coil (n = 27), (2) lumbar or thoracic spine MRI using only the in-table spine coils (n = 10), (3) hip MRI using a large flex coil (n = 8) and (4) upper- or lower-extremity MRI using small flex coils (n = 5). Following the MRI examinations, patients evaluated (1) sense of space, (2) noise level, (3) comfort, (4) coil comfort and (5) overall examination impression on a 5-point Likert-scale (range: 1= “much worse” to 5 = “much better”) using a questionnaire. Maximum noise levels of all performed imaging studies were measured in decibels (dB) by a sound level meter placed in the bore center. Results: Sense of space was perceived to be “better” or “much better” by 84% of patients for imaging examinations performed on the 0.55T MRI scanner system (mean score: 4.34 ± 0.75). Additionally, 84% of patients rated noise levels as “better” or “much better” when imaged on the low-field scanner system (mean score: 3.90 ± 0.61). Overall sensation during the imaging examination at 0.55T was rated as “better” or “much better” by 78% of patients (mean score: 3.96 ± 0.70). Quantitative assessment showed significantly reduced maximum noise levels for all 0.55T MRI studies, regardless of body region compared to 1.5T, i.e., brain MRI (83.8 ± 3.6 dB vs. 89.3 ± 5.4 dB; *p* = 0.04), spine MRI (83.7 ± 3.7 dB vs. 89.4 ± 2.6 dB; *p* = 0.004) and hip MRI (86.3 ± 5.0 dB vs. 89.1 ± 1.4 dB; *p* = 0.04). Conclusions: Patients perceived 0.55T new-generation low-field MRI to be more comfortable than conventional 1.5T MRI, given its larger bore opening and reduced noise levels during image acquisition. Therefore, new concepts regarding bore design and noise level reduction of MR scanner systems may help to reduce patient anxiety and improve well-being when undergoing MR imaging.

## 1. Introduction

Magnetic resonance imaging (MRI) has become indispensable in modern medicine. It is the gold-standard imaging modality for many soft-tissue and bone diseases [1,2,3,4]. However, while the diagnostic value of MRI is undisputed, patient acceptance and comfort during image acquisition require improvement. One of the main issues is the cramped patient position in the narrow MRI tube, explaining patients’ possible discomfort. Claustrophobia was reported to manifest in up to 15% of MRI examinations, depending on the imaged body region, in [5]. In addition to the unpleasant and possibly traumatic patient experience, discomfort results in the degradation of image quality due to motion artifacts, premature exam termination, lengthening of scan time due to the need to repeat sequences, and ultimately the need for sedation or anesthesia [6].

Several factors may be addressed to improve patient comfort, including the noise level of the examination, lighting, patient position in the scanner, coil design, duration of the examination, and type and size of the MRI borehole [7,8,9]. Accordingly, a previous study revealed the following patient requests for changes to this scanner design: lower noise levels, availability of music, and more overhead space, and especially the use of larger bore diameters [10]. In terms of patient interest, this development is favored by the renaissance of low-field MRI, since larger bores are technically easier and more cost-effective to realize at lower field strengths [11]. Another effect of lower field strengths is an expected lower acoustic noise level than with higher-field MRI.

Thus, the aim of this study was to evaluate the effects of a larger bore width of a new-generation low-field 80 cm bore scanner compared with a contemporary 1.5T machine with a 60 cm opening, specifically regarding the overall patient comfort, using a questionnaire. In addition, noise measurements were performed on both scanners.

## 2. Materials and Methods

This prospective study was approved by the Ethikkommission Nordwest- und Zentralschweiz ethics committee (BASEC 2021-00166). Informed consent was obtained from each patient.

### 2.1. Study Population

Between May 2022 and July 2022, patients were recruited to volunteer for a repeat examination at 0.55T following a routine 1.5T MRI examination. Inclusion criteria were signed patient consent and completion of the 1.5T MRI. Exclusion criteria were emergency examinations, 1.5T MRI including intravenous (i.v.) contrast media injection, critically ill and non-consenting patients. In total, 50 patients (28 male, 22 female, 66.2 ± 17.0 years) were prospectively enrolled, and underwent 0.55T within two hours maximum after the 1.5T MRI.

Four groups were defined depending on the imaged body region (Figure 1): (1) 27 patients received a cranial or cervical spine examination, (2) 10 patients received a lumbar or thoracic spine examination, (3) 8 patients received a hip examination and (4) 5 patients received an upper or lower extremity examination (3 patients with knee examination, 1 foot examination, 1 wrist examination). Indication for the cranial MRI was the clarification of a stroke. Patients with spinal examinations were scanned for disc degeneration and spinal or neuroforaminal stenosis. The hip examinations were all performed in patients with painful hip arthroplasty. The examinations of the knee joints, wrist and foot were performed for potential trauma sequelae.

### 2.2. MR Imaging

MRI was performed on a 0.55T scanner with a bore diameter of 80 cm (Siemens MAGNETOM FreeMax, Siemens Healthineers; Erlangen, Germany) and a 1.5T scanner with a diameter of 60 cm (Siemens MAGNETOM Avanto Fit, Siemens Healthineers; Erlangen, Germany; Figure 2). The technical specifications can be found in Table 1. Comparable protocols for 0.55T and 1.5T were used in all patients with identical sequences, adapted for the particular field strength. The only exception was the use of a Turbo-Inversion Recovery-Magnitude (TIRM) instead of a T2 DIXON as a fat-saturated coronary sequence at 0.55T. Table 2 provides the protocols for 0.55T and 1.5T for the three most imaged body regions (head, lumbar spine, hip). A head coil was used to examine the skull and cervical spine (1.5T: 32 channels, 0.55T: 8 channels). Examinations of the spine were performed using a spine array integrated into the table. Examinations of the hip and extremities were performed using 6-channel flex coils. Except for examination of the extremities, the patient position was head-first and supine in all cases. Knees and feet were examined in feet-first and supine position. The wrist was examined in a “superman position”.

### 2.3. Questionnaire

Following the MRI examinations, patients evaluated (1) sense of space, (2) noise level, (3) comfort, (4) coil comfort and (5) overall examination impression on a 5-point Likert-scale (range: 1 = “much worse” to 5 = “much better”).

### 2.4. Noise Measurements

Separate noise measurements were made on both scanners. These were performed without patients. The maximum noise levels in decibels (dB) of all sequences used for examinations of the skull, spine and hip were recorded. The microphone (DEM 200, Velleman Group; Gavere, Belgium) was located at the approximate position of the head of a person with a height of 175 cm, with the respective examination region in the isocenter. The measurements were repeated three times each.

### 2.5. Statistics

Mean and standard deviation were calculated for questionnaire ratings and noise levels in dB. For noise level as a continuous variable, we examined whether there was a significant difference between 0.55T and 1.5T. The Shapiro–Wilk test was used to test for normal distribution (α = 0.05). The comparison of the different groups was performed using the Wilcoxon rank sum test. A *p*-value < 0.05 was considered statistically significant. 

Pearson’s coefficient was calculated for the correlation between the overall sensation (question 5) of the study and each of the four other parameters (questions 1 to 4).

## 3. Results

All MRI examinations were performed without incident. No examination was prematurely terminated or interrupted. All questionnaires were completed in full by all 50 participants.

### 3.1. Questionnaire Results

An overview independent of body region can be found in Figure 3. The noise level of the 0.55T scanner with the 80 cm bore width was rated 3.9 ± 0.6, the comfort 3.7 ± 0.8, the sense of space 4.3 ± 0.8, the coil comfort 3.7 ± 0.6 and the overall examination impression 4.0 ± 0.8. A total of 5 patients (10%) rated the noise level as “much better” in the 0.55T scanner compared to the 1.5T scanner, 37 patients (74%) rated it as “better”, while 6 patients rated it as “equal” and 2 patients as “worse”. A total of 29 patients (58%) rated the comfort as “much better” (n = 7) or “better” (n = 22), while 21 patients rated it as “equal” (n = 18). Three patients rated 0.55T MRI examination comfort as “worse” compared with the 1.5T MRI; 42 (84%) patients rated the available space as “much better” (n = 25) or “better” (n = 17) in the 0.55T scanner compared to the 1.5T scanner, whereas 8 patients rated the space as “equal”.

There was a strong positive correlation between the overall sensation of comfort and the perception of the noise level (r = 0.66, *p* < *0*.001) as well as the perception of space (r = 0.69, *p* < 0.001). There was moderate positive correlation between the overall sensation (r = 0.39; *p* = 0.032) and the comfort lying down as well as the coil comfort (r = 0.39, *p* = 0.005). 

Depending on the body region, the following results were obtained for the rating of the criteria in the 0.55T compared with the 1.5T scanner: the noise level, sense of space, comfort and coil comfort were rated on average as “better” for all body regions (Figure 4 and Table 3). Twenty patients with examinations of the head and cervical spine rated the overall experience as “better” or “much better”, six as “equal” and only one as “worse”. Nine of ten patients with imaging of the thoracic or lumbar spine rated the examination overall as “better” or “much better”. All eight patients with hip examinations rated the overall experience as “better” or “much better”. Two patients with examinations of the extremities rated the examination as better, and three rated it as the same.

### 3.2. Measurement of the Noise Levels

Shapiro–Wilk test revealed the lack of a normal distribution in the noise measurements. The maximum noise level of the examinations was lower for examinations of the head (0.55T: 83.8 ± 3.6 dB vs. 1.5T: 89.3 ± 5.4 dB, *p* = 0.046), spine (0.55T: 83.7 ± 3.7 dB vs. 1.5T: 89.4 ± 2.6 dB, *p* = 0.004) and hip (0.55T: 86.3 ± 5.0 dB vs. 1.5T: 89.1 ± 1.4 dB, *p* = 0.402) for the 0.55T compared with the 1.5T scanner. The sequence with the highest maximum noise level for the head protocols used was lower for the 0.55T (SWI: 87.3 ± 0.6 dB) than for the 1.5T scanner (DWI: 94.7 ± 0.6 dB). The sequences with the highest maximum noise level were higher in the 0.55T than in the 1.5T scanner for the spine and hip protocol (Figure 5).

## 4. Discussion

The aim of our prospective study was to systematically assess patient comfort in a low-field scanner of the latest generation with 80 cm bore width in comparison to a 1.5T standard scanner with 60 cm bore width by means of a questionnaire. To our knowledge, this is the first study to analyze a latest-generation low-field scanner based on a scanner-to-scanner comparison.

Our study demonstrated that the new-generation 0.55T MRI could improve perceived patient comfort when compared with a contemporary 1.5T MRI in a prospective setting while maintaining acceptable diagnostic quality (Figure 6 and Figure 7). This was most recently demonstrated in a paper we published, in which we investigated the stroke-imaging performance of a 0.55T low-field MRI compared to a 1.5T scanner [12]. Similar initial results were additionally shown in a scanner–scanner comparison (0.55T vs. 1.5T) for spine imaging and hip-implant imaging, which are currently the subject of our research. 

Patients overwhelmingly rated the overall experience of the examination, sense of space and noise level as much better on the 0.55T scanner with 80 cm bore width compared to the 1.5T scanner with 60 cm bore width. This was true for all regions studied, i.e., both for examinations with and without a head coil. These results are in agreement with previously published results, which showed that a larger bore size was favored by most patients [10]. Improved coil design and volume reduction are also starting points that should make MRI more patient-friendly, and are desired on the patient side [9].

Coil comfort was also found to be higher in the 0.55T scanner for the tests where flex coils were used. This is likely explained by the field-strength-related coil geometry, which is more flexible at 0.55T. Overall, 78% of the patients rated the examination as better i.e., more comfortable, for the 0.55T compared with the 1.5T MRI. This is particularly remarkable since the examination at 0.55T took place immediately after the first routine examination, and the patients were informed that this examination was for comparison purposes only, without additional diagnostic benefit. As such, it can be assumed that there was lesser patient motivation to suppress urges for movement or endure discomfort to allow for optimum diagnostic image quality to be obtained.

Our quantitative findings also showed that, regardless of imaged body region and sequence, the 0.55T scanner was less noisy when compared with the 1.5T MRI. This is to be expected, since the primary source of loudness in MR investigations are vibrations of the gradient system. These are caused by Lorentz forces, which are proportional to the field strength. Various studies have shown that the noise level is one of the most important factors influencing patient comfort in MR imaging [13,14].

Overall, our study demonstrated that a significant increase in patient comfort can be achieved using a 0.55T scanner. The lower noise level, which was also subjectively rated as more comfortable by the patients, is directly attributable to the lower field strength. The new coil design, which is possible at lower field strengths due to the configuration, is also rated positively by the patients. A larger bore width is also possible at higher field strengths, but is physically more difficult to realize and therefore more expensive [15]. Further advantages outside the direct focus of this study are ecological and economic: low-field scanners of the latest generation promise lower acquisition and installation costs, and are less resource-intensive, which is attracting increasing interest socially and among patients [13,16]. However, increasing patient comfort in isolation is already of enormous importance, as it can avoid anesthesia, especially in children, and increase patient acceptance of MRI examinations [17,18].

There are several limitations that need to be discussed: first, we included only a small number of patients. Therefore, studies with a larger volume of participants are warranted. However, this is difficult for monetary and logistical reasons. Secondly, there was a selection bias because patients who are claustrophobic are less likely to undergo a second examination. In addition, there might be a further bias in this respect regarding the scan order: a second scan may be experienced positively or negatively for different psychological reasons. For example, the factor that the second scan is voluntary could lead to a more positive impression in the context of the study. Ideally, one could conduct a control experiment to investigate this effect. However, this was beyond the scope of our study and would ideally involve a separate study, and was therefore not conducted. Third, scanners with a bore width of 80 cm and 60 cm were compared. Thus, the evaluation of the space perception could once again be clearly in favor of the device with the larger bore width. Further studies should therefore also compare scanners with 70 cm bore and possibly other scanner designs, such as open-bore scanners. Fourth, when considering noise levels, it should be noted that the maximum noise level is not only dependent on the field strength, but also on the gradient design and, in particular, the sequences used. Therefore, a pure comparison in terms of field strength is only possible to a limited extent. Overall, it can be said that larger-scale prospective studies are needed to investigate the influence of bore size, noise level, and scanner geometry on patient comfort and patient acceptance. Ultimately, however, these comparisons are difficult. For example, examination times are longer at lower field strengths, which of course also has an impact on patient comfort, and open-bore scanners have disadvantages in terms of diagnostic power and are therefore not an equivalent substitute in every respect, which could necessitate the repetition of examinations [19].

In conclusion, this study showed that patients perceived 0.55T new-generation low-field MRI as more comfortable compared with conventional 1.5T MRI, given its larger bore opening and reduced noise levels during image acquisition. Thus, new-generation low-field MRI may increase patient acceptance and represent a viable alternative for anxious or claustrophobic patients, who could benefit from this new system.

## Figures and Tables

**Figure 1 jcm-11-06705-f001:**
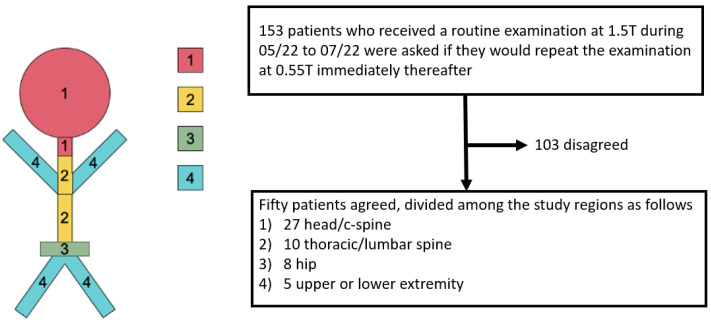
Overview of the patients included and their distribution among the study regions.

**Figure 2 jcm-11-06705-f002:**
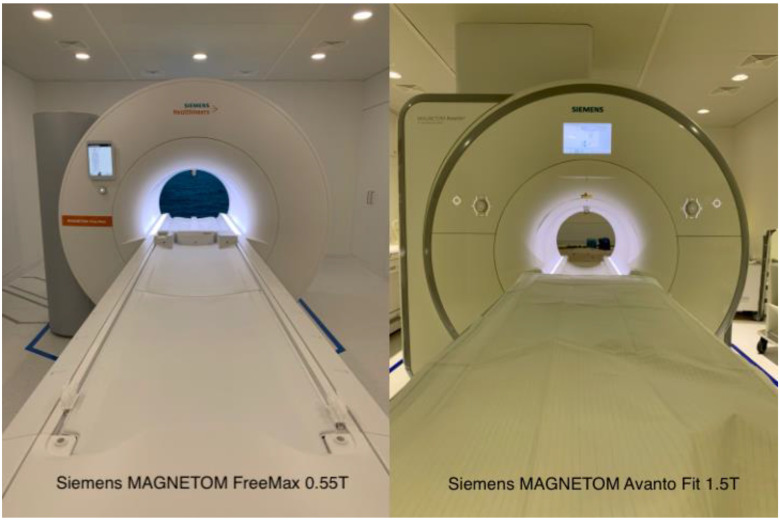
Sample pictures of the examination rooms of the Siemens MAGNETOM FreeMax 0.55T (**left side**) and Siemens MAGNETOM Avanto Fit 1.5T (**right side**).

**Figure 3 jcm-11-06705-f003:**
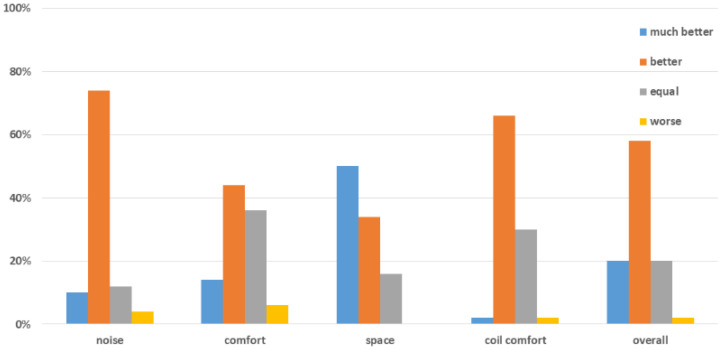
Overview of the patient ranking of the study at 0.55T compared to 1.5T in terms of noise level, lying comfort, coil comfort and overall sensation.

**Figure 4 jcm-11-06705-f004:**
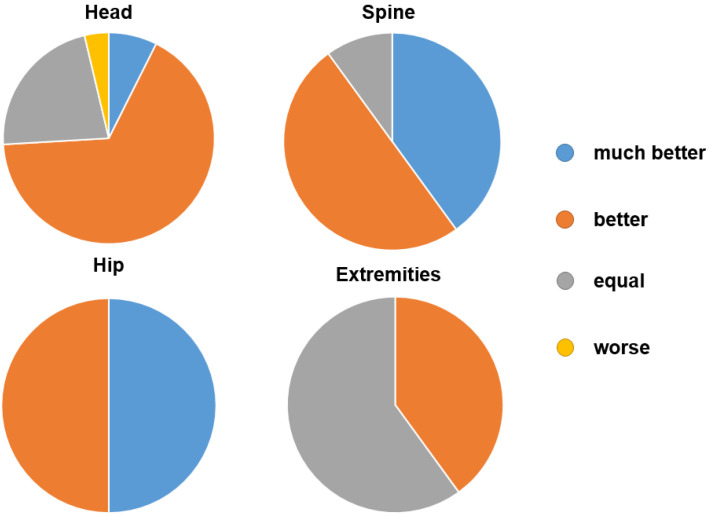
Breakdown of the overall examination score comparing 0.55T and 1.5T scanners, divided among the four different body regions.

**Figure 5 jcm-11-06705-f005:**
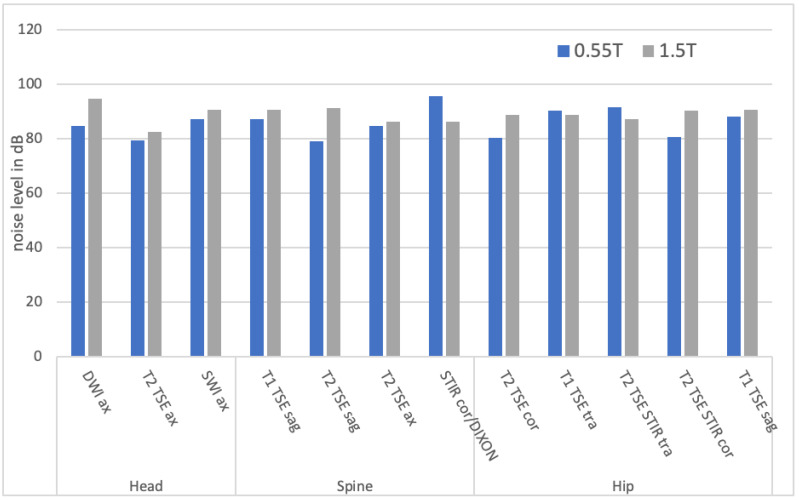
Maximum noise level in decibels (dB) for the individual sequences of the three body regions head, spine and hip.

**Figure 6 jcm-11-06705-f006:**
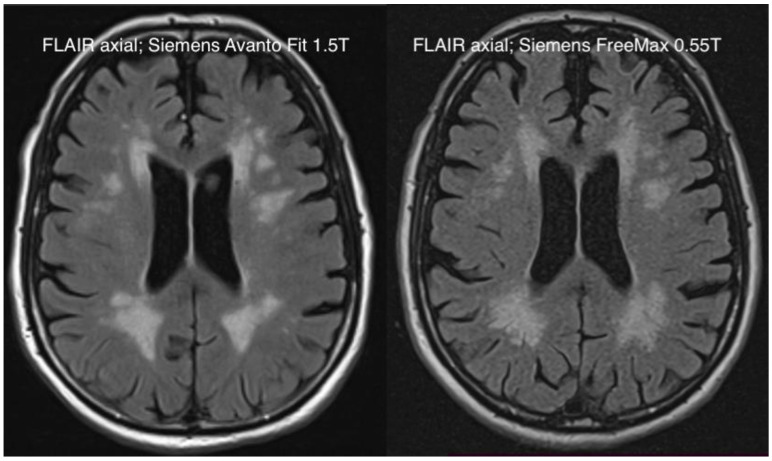
Exemplary comparison of an axial FLAIR sequence in the context of brain imaging with the Siemens Avanto FIT 1.5T (**left side**) and the Siemens Free Max 0.55T (**right side**).

**Figure 7 jcm-11-06705-f007:**
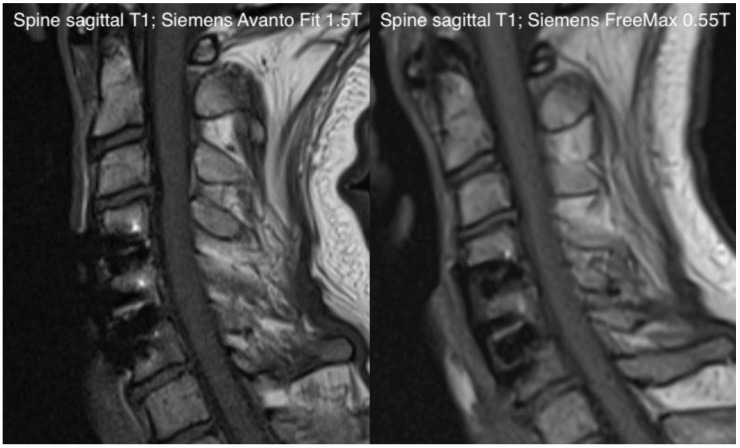
Exemplary comparison of a sagittal T1-sequence in the context of spine-imaging with the Siemens Avanto FIT 1.5T (**left side**) and the Siemens Free Max 0.55T (**right side**).

**Table 1 jcm-11-06705-t001:** Specifications of 0.55T and 1.5T MR scanner systems.

	Bore Width (cm)	Field Strength (T)	Gradient Amplitude (mT/m)	Slew Rate (T/m/s)
MAGNETOM Free Max	80	0.55	26	45
MAGNETOM Avanto FIT	60	1.50	45	200

**Table 2 jcm-11-06705-t002:** Protocols for 0.55T and 1.5T for the three most imaged body regions (head, lumbar spine, hip).

			Repitition Time (TR) (ms)	Echo time (TE) (ms)	Slice thickness (ST) (mm)	Resolution (mm^2^)	Field-of-view (FOV) (mm^2^)	Time of acquisition (TA) (min)
Lumbar spine	T1 Turbo spin echo (TSE) sagittal	1.5T	625	11	4	0.7 × 0.7	300 × 300	02:29
		0.55T	454	13	4	0.8 × 0.8	320 × 320	05:26
	T2 TSE sagittal	1.5T	3600	102	4	0.7 × 07	300 × 300	01:44
		0.55T	3500	99	4	0.8 × 0.8	320 × 320	03:34
	T2 TSE axial	1.5T	4210	107	4	0.5 × 0.5	200 × 200	04:40
		0.55T	5910	84	4	0.5 × 0.5	200 × 200	04:51
	T2 DIXON cor	1.5T	6630	90	5	0.8 × 0.8	300 × 300	05:00
Hip	TIRM cor	1.5T	4000	34	3.5	1.1 × 0.9	220 × 220	01:48
		0.55T	4440	35	3.5	1.6 × 1.1	220 × 220	02:48
	T2 TSE cor	1.5T	3400	73	3.5	0.6 × 0.5	220 × 220	01:25
		0.55T	3220	77	3.5	1.0 × 0.7	220 × 220	02:21
	TIRM axial	1.5T	5210	54	5	0.8 × 0.6	180 × 180	03:33
		0.55T	4260	25	5	1.1 × 0.9	220 × 220	04:58
	T1 TSE axial	1.5T	661	9.5	5	0.9 × 0.8	200 × 200	05:44
		0.55T	517	9.4	5	1 × 0.7	220 × 220	03:40
	T1 TSE sagittal	1.5T	596	8.5	5	0.9 × 0.7	220 × 220	02:32
		0.55T	517	9.4	5	1 × 0.7	220 × 220	03:40
Head	Fluid attenuated inversion recovery (FLAIR) axial	1.5T	8510	112	3	0.9 x 0.9	187 × 230	03:26
		0.55T	7780	96	3	1.3 × 1	209 × 230	05:28
	Susceptibility weighted imaging (SWI) axial	1.5T	48	40	3	1.1 × 0.96	194 × 230	02:17
		0.55T	172	100	3	0.9 x 0.8	201 × 230	02:23
	Diffusion weighted imaging (DWI) (echo-planar imaging(EPI)) axial	1.5T	6200	103	3	1.4 × 1.4	220 × 220	02:04
		0.55T	7400	102	3	1.7 × 1.7	230 × 230	04:35

**Table 3 jcm-11-06705-t003:** Results of the patient evaluation with regard to noise, comfort, space perception, coil comfort and the overall experience (mean ± standard deviation) depending on the examined body region.

	Noise	Comfort	Space	Coil Comfort	Overall
Head/C-Spine	3.9 ± 0.7	3.9 ± 0.7	4.4 ± 0.8	3.6 ± 0.6	3.8 ± 0.6
T-/L-Spine	3.9 ± 0.6	3.6 ± 1.0	4.3 ± 0.7	3.5 ± 0.5	4.3 ± 0.7
Hip	4.1 ± 0.3	3.4 ± 0.5	4.8 ± 0.5	3.9 ± 0.4	4.5 ± 0.5
Extremities	3.8 ± 0.5	2.8 ± 0.5	3.4 ± 0.6	4.0 ± 0.0	3.4 ± 0.6

## Data Availability

The data presented in this study are available on request from the corresponding author.

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
