# Peer review of "More Space, Less Noise—New-generation Low-Field Magnetic Resonance Imaging Systems Can Improve Patient Comfort: A Prospective 0.55T–1.5T-Scanner Comparison"

_jcm, 2022, doi:10.3390/jcm11226705_

Round 1

Reviewer 1 Report

Summary:
The authors present a well conducted study, while the materials & methods seem to be sound. Furthermore, this is a well written manuscript which deals with the rebirth of an increasingly topical issue. New generation Low-Field Magnetic Resonance Imaging Systems seems to undergo a renaissance. The manuscript opens the floor for new concepts regarding the bore design and noise level reduction further reducing patient anxiety, still representing one of the major contributors for poor imaging quality and therefore uncertainty in our radiological reports. Therefore, I would like to encourage the authors to further dedicate themselves to this scientific topic area.

Specific Comments
Abstract / Keypoints
- There is a clear connection between Objectives, Purpose and Conclusion

Keywords:
- I would add also MRI

Informed Consent:

-Not nessecary.

Introduction:
No comments

Materials and methods:
-Overall, well-constructed with a well defined search strategy and study selection. Furthermore, the statistical analysis seems to be profound.

Results:
-No comments.

Discussion
-No comments

-Appropriate Limitation paragraph

Figures and Tables
-  It is a pity that the authors have not included any illustrations in such a successful manuscript. Both a comparative image showing the image quality of the examinations and an image of the examination room with the low-field scanner would give the reader a better insight into the marteria.

Language
-Although well written it is always advisable to let the manuscript be checked by a professional are native speaker.

Author Response

Please see attachment (Point-by-points answers). 

Dear Reviewer,

thank you very much for your constructive reviews.

We have now addressed the comments and suggestions for improvement in the revised manuscript and implemented them. In addition, we have provided a detailed point-by-point explanation of the individual suggestions for improvement and explained our approach in it. We hope that the changes are now to your complete satisfaction. From our point of view, this has enabled us to improve our manuscript once again. We would like to thank you for this and hope that it now meets the criteria for publication.

If you have any further questions or criticism, please do not hesitate to contact us.

Thank you very much.

Best regards,

Thilo Rusche, MD, on behalf of all authors

Reviewer 2 Report

In the paper “More space, less noise – new-generation low-field magnetic resonance imaging systems can improve patient comfort: A prospective 0.55T-1.5T-scanner comparison”, Dr. Rusche et al compares the patient comfort in two different MRI scanners of different bore size and different magnetic field strength. The study is prospective, enrolling n=50 patients (of 153 asked to participate), who were asked to repeat an MRI scan in the 1.5T scanner also in the 0.55T scanner, and compare the experience in noise, comfort, space, coil comfort, and overall on a 5-point scale from “much worse” to “much better”. The wider-bore 0.55T scanner was generally preferred by the patients. A better patient experience may increase patient acceptance of an MRI scan.

Overall, the paper is to the point and presents clear findings, and suitably discusses these findings and their possible limitations. The following comments are about a few issues where the paper could still be strengthened.

1. There might be an effect of the order of the scans. For example, a patient might think “This second time I know what a scan is like, and I did this voluntarily”, leading to a more positive evaluation of the second scan, simply because it is the second. Conversely, a patient might think “This is still noisy, and the space is once again cramped – why did I say yes to a second scan?”, which could lead to a more negative evaluation of the second scan, simply for being second. Ideally, a control experiment could be made, scanning a group of patients in two similar scanners, to evaluate such an effect. Practically, such a control study would be a lot of work in itself, and likely not a realistic option. But the authors should at least discuss the possibility in the limitations section.

2. The comparison only includes the patients’ experience. It would be interesting to hear how the clinicians felt about the images from a 0.55T scanner in comparison to a 1.5T scanner. This could be done based on the available data from the study (thereby being more realistic than the control study hinted at in the previous comment), but might also be a study in itself. At least some overall indication should be given here – whatever the patients think about a new scanner, it is only clinically useful if it can produce clinically usable scans.

3. The statistics section 2.5 mentions that the data were tested for the normality (with the Shapiro-Wilk test), but the results of the test does not seem to be reported in the Results section? The choice of Wilcoxon’s signed rank test (rather than Student’s t-test) for comparison hints that the data were not normally distributed, but guessing should not be needed. Please write a few words on this.

4. In Figure 2, the legend is incomplete, telling only what dark blue and light blue means, not what gray and red means.

5. Figure 3 has a complete legend, but with different colours than in Figure 2. Why not use the same colour scale in Figure 2 and Figure 3?

6. In section 3.1, the means and SD are reported with two decimal figures. It is suggested to present only a single decimal figure, as the SD clearly indicates that the second decimal figure caries very little information. E.g. the results 3.90 +/- 0.61 and 3.66 +/- 0.80 could be presented as 3.9 +/- 0.6 and 3.7 +/- 0.8. This is easier to read and has not lost any practically useful information. If this suggestion is used, the results in Table 3 and section 3.2 should be similarly rounded. Talking of Table 3, please be consistent in the number of decimals, also when the last decimal happens to be 0.

Author Response

Please see the attachment (Point-by-point answers).

Dear Reviewer,

thank you very much for your constructive reviews.

We have now addressed the comments and suggestions for improvement in the revised manuscript and implemented them. In addition, we have provided a detailed point-by-point explanation of the individual suggestions for improvement and explained our approach in it. We hope that the changes are now to your complete satisfaction. From our point of view, this has enabled us to improve our manuscript once again. We would like to thank you for this and hope that it now meets the criteria for publication.

If you have any further questions or criticism, please do not hesitate to contact us.

Thank you very much.

Best regards,

Thilo Rusche, MD, on behalf of all authors
